# Rhododenol Activates Melanocytes and Induces Morphological Alteration at Sub-Cytotoxic Levels

**DOI:** 10.3390/ijms20225665

**Published:** 2019-11-12

**Authors:** Minjeong Kim, Chang-Seok Lee, Kyung-Min Lim

**Affiliations:** 1College of Pharmacy, Ewha Womans University, Seoul 03760, Korea; tyndall@ewha.ac.kr; 2Department of Beauty and Cosmetic Science, College of Health Science, Eulji University, Seongnam-si 13135, Korea

**Keywords:** rhododenol, melanocytes, Melanoderm™, leukoderma, cytoskeleton, morphology

## Abstract

Rhododenol (RD), a whitening cosmetic ingredient, was withdrawn from the market due to RD-induced leukoderma (RIL). While many attempts have been made to clarify the mechanism underlying RIL, RIL has not been fully understood yet. Indeed, affected subjects showed uneven skin pigmentation, but the features are different from vitiligo, a skin hypopigmentary disorder, alluding to events more complex than simple melanocyte cytotoxicity. Here, we discovered that rhododenol treatment reduced the number of melanocytes in a pigmented 3D human skin model, Melanoderm™, confirming the melanocyte toxicity of RD. Of note, melanocytes that survived in the RD treated tissues exhibited altered morphology, such as extended dendrites and increased cell sizes. Consistently with this, sub-cytotoxic level of RD increased cell size and elongated dendrites in B16 melanoma cells. Morphological changes of B16 cells were further confirmed in the immunocytochemistry of treated cells for actin and tubulin. Even more provoking, RD up-regulated the expression of tyrosinase and TRP1 in the survived B16 cells. Evaluation of mRNA expression of cytoskeletal proteins suggests that RD altered the cytoskeletal dynamic favoring cell size expansion and melanosome maturation. Collectively, these results suggest that RD not only induces cytotoxicity in melanocytes but also can lead to a profound perturbation of melanocyte integrity even at sub-cytotoxic levels.

## 1. Introduction

Rhododenol (4-(4-hydroxyphenyl)-2-butanol, RD) is a naturally occurring phenolic compound found in plants, such as *Acer nikoense* and *Betula platyphylla* [1]. Since 2008, a cosmetics company in Tokyo, Japan, has used the racemic form of RD (RS-RD) as a whitening cosmetic ingredient [2]. In July 2013, a large number of consumers of RD-containing cosmetics complained about leukoderma in their face, neck, and hands. Subsequently, cosmetics containing RD were immediately recalled and withdrawn from the market. However, 19,605 (as of October 2016) among approximately 800,000 RD users have suffered RD-induced leukoderma (RIL), which amounts to a 2 to 2.5% incidence rate [3].

Skin biopsies taken from the decolorized lesions of affected subjects had few or no melanocytes compared to normal skin [4]. On the other hand, RD had no cytotoxic effect on keratinocytes and fibroblasts [5]. These data strongly suggest that RD has a direct and toxic effect selectively on melanocytes and induces chemical-induced leukoderma. Supporting this, several studies demonstrated that RD induces cytotoxicity in melanocytes through oxidative stress for which a comprehensive review has been recently published [6]. Tokura et al. reported that RD is catalyzed by tyrosinase to produce toxic prooxidant metabolites, such as RD-cyclic catechol [2]. Previously, we also demonstrated that RD generates reactive oxygen species (ROS), induces DNA damages, and impairs normal cell proliferation [7] in melanocytes. In another study, it was demonstrated that RD induces endoplasmic reticulum (ER) stress in a tyrosinase-dependent manner, which further activates the apoptotic pathway [5].

Interestingly, Tsutsumi et al. recently reported that RIL exhibits feature different from vitiligo, a hypopigmentary disease [8]. While vacuolar changes, melanophage, perifollicular lymphocyte infiltration, and loss of melanin were commonly observed in both conditions, RIL distinctively exhibited remnant melanocytes in the lesion, with heterogeneous melanization, degenerated melanosomes and intact cell organelles, reflecting the involvement of more complex events in RIL.

Melanocytes are melanin-producing neural-crest derived cells located in the lowest layer of the epidermis of the skin [9]. Factors regulating skin pigmentation like α-MSH or ACTH bind to the melanocortin-1 receptor (MC1R) and activate intracellular adenylate cyclase. This increases the concentration of cyclic adenosine monophosphate (cAMP), which upregulates tyrosinase, the rate-limiting enzyme for melanin biosynthesis, along with TRP1 and TRP2, through the cell signaling pathway of protein kinase A (PKA) [10]. Synthesis of melanin occurs in a special organelle called melanosomes, which are carried along the cytoskeleton by tubulins and actin-dependent motor proteins like Rab27a, melanophilin, and myosin Va, towards the pericellular melanocyte dendrites [11,12].

Ultimately, melanosomes are transferred to keratinocytes proliferating and differentiating outwards from the basal layer [13,14]. Even without evident cytotoxicity, impairment of melanin synthesis and melanosome transfer may affect the normal melanocyte homeostasis.

Reflecting on previous mechanistic studies on RIL, cytotoxic levels of RD (≥0.5 mM, ≥~90 μg/mL in monolayer cells in vitro) [15] were mainly employed to study the toxicity of RD to melanocytes, but the effects of sub-cytotoxic levels of RD (≤0.25 mM or ≤50 μg/mL) were not fully addressed. Here, we aimed to investigate whether sub-cytotoxic levels of RD can affect melanocytes homeostasis. To address long-term use conditions of whitening cosmetics in more in vivo-like conditions, 3D pigment human epidermis model, Melanoderm™ was used, as well as B16 melanoma cells, for the imaging and gene expression study. Morphological examinations of melanocytes and melanoma cells were conducted using immunohistochemistry and immunocytochemistry, along with qPCR for melanosome proteins and cytoskeletal proteins to understand RIL.

## 2. Results

### 2.1. Effects of RD Treatment on A Pigmented Human Skin Model, Melanoderm™

Artificial human skin models are widely used to examine the anti-pigmentary effects and safety of cosmetic ingredients in an in vivo-like condition [16]. We examined whether RD could attenuate melanin synthesis in an artificial pigmented skin model, Melanoderm™. RD was treated at 0.25, 0.5 and 0.8% (2500, 5000, and 8000 μg/mL, respectively, by placing about 100 fold margin from the cytotoxic concentration, 90 μg/mL and sub-cytotoxic concentration, 25 μg/mL and 50 μg/mL observed in 2D cell culture to take incomplete skin penetration of RD through 3D human epidermis into consideration) every other day for 17 days and color alteration of the skin tissue was photographed. RD brightened the color of the skin tissue and significantly changed the degree of pigmentation, as represented by significantly reduced ∆L at 0.5% and 0.8% (Figure 1B,C). However, at the same time, cytotoxicity of RD to the tissues was confirmed by water-soluble tetrazolium-1 (WST-1) assay, which showed a significant decrease of viability in RD 0.5% and 0.8% compared to the control group (Figure 1D), while at 0.25%, only minimal effects were observed.

The effects of RD on the melanocyte of Melanoderm™ were further investigated through histological examinations after hematoxylin and eosin (H&E) staining or Fontana-Masson (F-M) staining (Figure 2). Reduced number of melanocytes was observed at RD 0.5%. Interestingly, at 0.25% where RD showed only minimal effects on cell viability, melanocytes were also markedly decreased. Remarkably, F-M staining demonstrated that melanocytes remaining after RD treatment, showed hypertrophy (increases in cell size), dendrite formation and increased melanization compared to the control group, suggesting that morphological changes may proceed the onset of cytotoxicity.

### 2.2. Morphometric Analysis of B16 Melanoma Cells Following Exposure to Sub-Cytotoxic Levels of RD

Using sub-cytotoxic levels of RD on B16 melanoma cells, we verified morphological changes of melanocytes. As expected, RD stimulated melanocytes showed extended dendrites and increased cell size in comparison to the control (Figure 3A). RD significantly increased cell sizes at both 25 and 50 μg/mL at 24 h, while RD at 50 μg/mL elongated dendrites at 48 h exposure (Figure 3B,C), confirming that RD can induce morphological changes of melanocytes at sub-cytotoxic levels.

### 2.3. Immunocytochemistry to Examine Cell Size, Dendrite Formation, and Melanosomes After RD Exposure

We further investigated the morphological changes of B16 cells using immunocytochemistry for actin (red) and tubulin (green), key cytoskeletal proteins. As a result, the significant changes in size were observed at the RD treated group (25 and 50 μg/mL, Figure 4A,B) in line with the results above (Figure 3). Interestingly, an extension of tubulin (red in Figure 4A,B) was evident, while actin was relatively unaffected.

In Figure 1, we observed that remaining melanocytes showed increased melanization. To further investigate it, we conducted immunocytochemistry for a melanosome marker, TRP1 (Figure 5). Remarkably, RD treated B16 cells showed more intense TRP1 signal (red), along with increased distribution, which could confirm the Melanoderm™ data.

### 2.4. Expression of Genes and Proteins Related to The Cytoskeleton and Melanosomes

To confirm the morphological changes observed in RD-treated B16 melanoma cells, we conducted qPCR analysis (Figure 6). Increased levels of tyrosinase, TRP1 and TRP2 were observed, reflecting that melanogenesis was activated in remnant melanocytes after RD treatment (Figure 6A). Interestingly, while the expression of cytoskeletal proteins like tubulin or actin was unaffected (Figure 6B), actin-related protein complex 2 (Arpc2) and Arpc4 (Actin-related protein 2/3 sub2, 4) were upregulated. Moreover, Rab27, a key protein for melanosome transfer, was upregulated, which is in line with increased melanogenesis.

Upregulations of TRP1, b-tubulin, and Rab27A were further confirmed by western blot analysis, which showed dose-dependent increases in the protein level following RD exposure (Figure 7).

## 3. Discussion

In this study, we demonstrated that RD at sub-cytotoxic levels induces morphological changes of melanocytes in Melanderm™ and B16 melanoma cells, which accompanies cytoskeletal changes and increased melanization. Melanogenesis is a complex physiological mechanism involving various paracrine factors. Skin cells, such as melanocytes, keratinocytes, and fibroblasts, communicate with one another through secreted regulators, thereby regulating the melanocytes’ bio-functions [17]. Cytoskeleton related genes that increased following RD exposure include tubulin, and Arp2/3 complex 2, 4. In addition, melanogenic enzymes, tyrosinase, and TRP1 increased in the remnant melanocytes after exposure to sub-cytotoxic levels of RD. Melanosome transfer appeared to be activated as well, as could be determined by increased expression of Rab27. Western blot analysis further confirmed these changes. Taken together, these results may provide an important clue to explain the histological features of remnant melanocytes in the lesion of human RIL which exhibits heterogeneous melanization with intact cell organelles [8].

Transfer of melanosomes is dependent on actin and is mediated through the formation of a tripartite protein complex composed of Rab 27a-Slac 2-myosin Va (MyoVa). Transport of melanosome is normally initiated by binding of an effector protein, Rab27A, a small GTPase protein. Rab 27A transports melanosomes from the perinuclear region of melanocytes to the cell membrane [18]. RAB27A gene is regulated by Microphthalmia-associated transcription factor [19,20], a common transcription factor upregulating tyrosinase, and TRP1, and TRP2, which is in line with our results showing increased expression of tyrosinase and TRP1. Similarly, regarding the melanocyte activation mechanisms and therapeutic topical treatment of SLs (solar lentigos), it is suggested that treatment of a tyrosinase inhibitor is a desirable treatment for SLs [21]. Activation of melanogenesis and melanosome transfer may enable melanocytes to produce more melanin and distribute melanosomes through forming dendrites, which may explain the activated melanocytes in RD-treated Melanoderm™.

Cytoskeleton is a major component of intracellular microtubules and functions in many processes, including structural support, intracellular transport, and DNA separation [22]. In order to form microtubules, α- and β-tubulin dimers bind to GTP and assemble into (+) ends of microtubules, while in GTP-bound state [23]. The β-tubulin subunit is exposed to the microtubule plus end, whereas the α-tubulin subunit is exposed to the minus end. After dimers are incorporated into microtubules, GTP molecules bound to the β-tubulin subunit are finally hydrolyzed to GDP via inter-dimer contacts along microtubule protofilaments [24]. Meanwhile, Arp2/3 complex, a seven-subunit protein complex, plays an important role in the regulation of actin cytoskeleton. It is a major component of the actin cytoskeleton and is found in eukaryotic cells containing most of the actin cytoskeleton [25]. Arp2/3 complex binds to the side of the existing filament and starts growing new filaments at an angle of 70 degrees. Branched actin networks are produced as a result of nucleation of new filaments, which is key for the rearrangement of the actin cytoskeleton in the processes, such as cell migration, phagocytosis, and intracellular motility of lipid vesicles [26].

In this study, RD treated B16 cells showed elongated dendrites. Melanocyte dendrites contain actin and microtubule and transport melanosomes to the tip of dendrites [27]. We demonstrated that β-tubulin and Arp2/3 complex were upregulated in the RD-treated B16 cells, which is in line with the increased cell size and elongated dendrites. Kawasaki et al. demonstrated that overexpression of MITF leads to increased dendricity of melanophores, as well as melanosome maturation [28], suggesting that RD-induced melanization and morphological changes may share the mechanism although further studies are necessary to confirm it.

Melanocytes deliver melanin to adjacent keratinocytes in the form of a melanosome package when activated through increased melanin synthesis and dendrite formation [29,30]. Melanocytes are vulnerable to toxicity, and when exposed to cytotoxic chemicals, they will die and result in hypopigmentary disorders like leukoderma or vitiligo [31]. However, the consequence may differ in some times and hyperpigmentation or uneven pigmentation may occur [32]. It is well established that RD induces leukoderma in human. This is most probably because dermal melanocytes are exposed to cytotoxic level of RD, which would lead to cell death and resultant leukoderma. It appears that whitening effects of RD is closely associated with cytotoxicity, and whitening dose of RD is converged with cytotoxic dose [33]. However, it is not known what effects the sub-cytotoxic levels of RD can inflict on dermal melanocytes. Focusing on the recent histological study of RD-induced leukoderma patients revealing the paradoxical melanization of remnant melanocytes [8], we hypothesized that sub-cytotoxic level of RD may cause effects on melanocytes distinct from simple cytotoxicity. As shown in our study, sub-cytotoxic levels of RD induced the activation of melanocytes and cytoskeletal changes as can be evidently supported by strong melanization and dendrite formation of remnant melanocytes in Melanoderm™ following the exposure to RD. Incidentally, Kim et al. reported that H_2_O_2_ induced melanogenesis through upregulating the expression of PAH, TYR, and MITF, and the phosphorylation of CREB in B16F10 and SK-Mel-2 cells [34]. RD induces oxidative stress in melanocytes and even sub-cytotoxic levels of RD perturb antioxidant homeostasis [35], suggesting that RD-induced ROS may be attributable to RD-induced melanocyte activation. Along with the cytotoxic effects of high dose RD, this melanization would result in uneven pigmentation in the affected patients.

In conclusion, we demonstrated that sub-cytotoxic levels of RD induce a profound perturbation of melanocyte homeostasis, resulting in the activation of melanogenesis and morphological changes. We believe that our results will help to understand RD induced adverse effects and expand the knowledge on the melanocyte toxicity.

## 4. Materials and Methods

### 4.1. Chemicals

Rhododenol (4-(4-hydroxyphenyl)-2-butanol) with the purity >90% was synthesized and checked for purity by Amorepacific R&D center (Yongin-si, Korea). WST-1 (4-[3-(indophenyl)-2-(4-nitrophenyl)-2H-5-tetrazolio]-1,3-benzene disulfonate) (Roche, Indianapolis, IN, USA), 4% phosphate-buffered formalin (Thermo Scientific, Waltham, MA, USA), nuclear fast red (60700, Fluka, Ronkonkoma, NY, USA), TRIzol reagent (Invitrogen, Carlsbad, CA, USA).

### 4.2. Human Epidermal 3D Skin Model, Melanoderm™

Melanoderm™ (MatTek, Ashland, MA, USA) is an artificial human epidermis consisting of normal human-derived epidermal keratinocytes and normal human-derived melanocytes (NHM) that exhibit uniform and highly reproducible morphological and ultrastructural characteristics. Briefly, Melanoderm™ was pre-incubated for 24 h, then treated with the indicated compounds every other day for 17 days. The ∆L value was analyzed with Adobe Photoshop CC 2015 software (San Jose, CA, USA).

### 4.3. Cell Viability Assay (WST-1)

WST-1 (4-[3-(indophenyl)-2-(4-nitrophenyl)-2*H*-5-tetrazolio]-1,3-benzene disulfonate) (Roche, Indianapolis, IN, USA) solution was used to investigate cell viability [36]. RD treated Melanoderm™ were incubated with 300 μL of WST-1 solution for 3 h at 37 °C. 5% CO_2_ in the dark. Two hundred microliters of supernatant was transferred into each well of a 96-well plate, and absorbance was measured at 450 nm. All measurements were performed in triplicate.

### 4.4. Histological Analysis

For the histological examination, all samples were fixed in 4% phosphate-buffered formalin (PFA) for 24 h with gentle shaking. Fixed samples were paraffin-embedded and cut into a 5-μm section using microtome (RM2335, Leica, Wetzlar, Germany). Hematoxylin and eosin staining (H&E) proceeded after 1 day of sectioning. For H&E staining, paraffin sections were deparaffinized and then hydrated in a descending grade of ethanol. Next, sections were stained with 0.1% Mayer’s hematoxylin for 10 min and 0.5% eosin in 95% EtOH. After staining with H&E, the washing steps were immediately and sequentially proceeded as follows: dip in distilled H_2_O until eosin stops streaking, dip in 50% EtOH 10 times, dip in 70% EtOH 10 times, incubate in 95% EtOH for 30 s and 100% EtOH for 1 min. Then, samples were covered with a mounting solution (6769007, Thermo Scientific, Waltham, MA, USA) and examined under the light microscope (BX43, OLYMPUS, Tokyo, Japan).

### 4.5. Fontana-Masson’s Argentaffin Staining

For F-M’s argentaffin staining, paraffin sections were incubated with ammoniacal silver nitrate for 1 h at 60 °C, followed two washed in distilled water. To develop color, samples were incubated with 0.2% gold chloride working solution for 10 min and immediately rinsed in distilled water 10 times. After final washes, samples were incubated with 5% sodium thiosulfate for 5 min to fix silver and rinsed with running water for 1 min. After the fixation of silver, nuclear fast red (60700, Fluka, Ronkonkoma, NY, USA) was used for counterstaining of samples and rinsed with running water for 1 min.

### 4.6. Cell Culture

The B16F10 cell line from C57BL/6 mice was purchased from ATCC (Manassas, VA, USA). Cells were maintained in standard culture conditions, Dulbecco’s Modified Eagle’s Medium (DMEM) supplemented with antibiotics (100 U/mL of penicillin A and 100 U/mL of streptomycin) and 10% fetal bovine serum (FBS) at 37 °C in a humidified atmosphere containing 5% CO_2_. At 80% cell confluence, adherent cells were detached with a solution of trypsin (Hyclone, South Logan, UT, USA).

### 4.7. Morphometric Analysis of Melanocyte Alteration

After treatment, media was removed and cells were washed twice with phosphate-buffered saline (PBS). Melanocytes were observed and photographed under a phase-contrast optical microscope with the magnification of 400 (X400, eclipse Ts2R, Nikon, Tokyo, Japan). For the counting of dendrites, elongated extensions from the cell body exceeding 50 pixels in ImageJ software (NIH, Bethesda, MD, USA) were acknowledged as a dendrite. For the measurement of cell size, contour of the cells were drawn in a polygon and the area was calculated in ImageJ software.

### 4.8. Immunocytochemical Reagents

Anti-beta actin antibody [primary antibodies (Abs)] were obtained from Abcam (1:200, Cambridge, UK). The secondary Ab was an Alexa Fluor^®^ 488-conjugated goat anti-rabbit IgG H&L (1:200, Abcam, Cambridge, UK). Anti-tubulin antibody (primary antibodies (Abs)) was conjugated with alexa fluor594 (1:150, Abcam, Cambridge, UK). Anti-TRP1 antibody (primary antibodies (Abs)) were obtained from Abcam (1:200, Cambridge, UK). The secondary Ab was an alexa fluor594-conjugated Goat Anti-Mouse IgG H&L (1:200, Abcam, Cambridge, UK).

### 4.9. Immunocytochemistry

Cells (3 × 10^4^/well) in complete medium were plated out on coverslips and incubated at 37 °C with 5% CO_2_ for 1–2 days until they were approximately 40% confluent. Cultured cells were rinsed with PBS and fixed for 15 min in 4% paraformaldehyde at room temperature (RT). After fixation with paraformaldehyde, cells were permeabilized with 0.25% Triton X-100 for 10 min. Cells were washed three times in PBS, blocked for 30 min in 1% bovine albumin (BSA) dissolved in PBS-0.1% Tween (PBST), and incubated at RT with the primary antibody in 1% BSA. Coverslips were rinsed twice in PBS, incubated for 1 h in secondary Ab in 1% BSA at RT, rinsed three times in PBS, and nuclei were counterstained with 4,6-diamidino-2-phenylindole (DAPI). Cells were washed three times in PBS, mounted on slides with prolong gold antifade reagent (Invitrogen, Carlsbad, CA, USA).

### 4.10. RNA Sample Preparation

B16F10 cells treated with RD for 24 and 48 h were collected and washed once with phosphate-buffered saline (PBS), and the total ribonucleic acid (RNA) was extracted with TRIzol reagent (Invitrogen, Carlsbad, CA, USA) according to the manufacturer’s protocol. After the addition of chloroform, samples were centrifuged at 12,000 rpm for 10 min. The aqueous phase was mixed with isopropanol and RNA pellets were collected by centrifugation (12,000 rpm, 15 min, 4 °C). RNA pellets were washed with 70% ethanol and dissolved in RNase-free, DEPC (diethylpyrocarbonate)-treated water (Waltham, MA, USA). The RNA yield was estimated by determining the optical density at 260 nm with a NanoDrop 1000 spectrophotometer (NanoDrop Technologies, Inc., Wilmington, DE, USA).

### 4.11. Real-Time PCR

Relative expression levels of mRNAs were measured by quantitative real-time PCR. Total RNA, extracted from melanocytes treated with RD, was used to synthesize cDNA using a pre-master mix with oligo dT (Bioepis, Seoul, Korea). SYBR Green PCR master mix and a StepOnePlusTM Real-time PCR machine (Applied Biosystems, Warrington, UK) were used in each reaction. Primers were purchased from Cosmogenetech (Seoul, Korea). The sequence of primers was as follows: forward α-tubulin, 5′-GCC CAA CCT ACA CCA ACC TT-3′; reverse α-tubulin, 5′-GGA AGT GGA TGC GAG GGT AG-3′; forward β-tubulin, 5′-GCT GGA CCG AAT CTC TGT GT-3′; reverse β-tubulin, 5′-CCC AGA CTG ACC GAA AAC GA-3′; forward mouse Actb, 5′-TGT TAC CAA CTG GGA CGA CA-3′; reverse mouse Actb, 5′-GGG GTG TTG AAG GTC TCA AA-3′; forward Tyrosinase, 5′-GGG CCC AAA TTG TAC AGA GA-3′; reverse Tyrosinase, 5′-ATG GGT GTT GAC CCA TTG TT-3′; forward TRP1, 5′-CTT TCT CCC TTC CTT ACT GG-3′; reverse TRP1, 5′-TCG TAC TCT TCC AAG GAT TCA-3′; forward TRP2, 5′-TTA TAT CCT TCG AAA CCA GGA-3′; reverse TRP2, 5′-GGG AAT GGA TAT TCC GTC TTA-3′ forward Rab27A, 5′-CCA GAG GGC AGT GAA AGA GG-3′; reverse Rab27A, 5′-CCG CTT CAT GAT CAG GTC CA-3′; forward PTK2, 5′-CCT CAG CTA GTG ACG TGT GG-3′; reverse PTK2, 5′-GGG AGG ACA ATT TGG AGG CA-3′; forward Arpc2, 5′-GCC GGA AAC AAA CCA GAA GC-3′; reverse Arpc2, 5′-CTC ATC AGC TCC ATG TGC CT-3′; forward Arpc4, 5′-CGA AGG AAA CCT GTG GAG GG-3′; reverse Arpc4, 5′-GGC ATT GAC CGA CAG CTT CA-3′. Cycling parameters were 50 °C for 2 min, 95 °C for 10 min, 40 cycles of 95 °C 15 s, and 50 °C 1 min.

### 4.12. Western Blot

B16F10 cells were washed after 48-h exposure to RD at 25 μg/mL and 50 μg/mL. Then, cells were homogenized in RIPA buffer (Sigma, St. Louis, MO, USA) containing 1% protease inhibitor cocktail and a phosphatase inhibitor cocktail. The homogenate was centrifuged (12,000 rpm, 10 min) and the supernatant was collected. The protein concentration was measured and an aliquot (15.6 μg protein) was subjected to 10% sodium dodecyl sulfate-polyacrylamide gel electrophoresis (SDS-PAGE), and the fractionated proteins were then transferred to a nitrocellulose membrane. For immunoblotting, the following primary antibodies were used: rabbit anti-TRP1 monoclonal antibody (1:20000 dilution; Santa Cruz, CA, USA), mouse anti-β-Tubulin monoclonal antibody (1:5000 dilution; Merck, Darmstadt, Germany) and rabbit anti Rab27A monoclonal antibody (1:1000 dilution; Santa Cruz, CA, USA) after incubation with HRP-conjugated secondary antibodies (KPL, Gaithersburg, MD, USA), the immunoreactive bands were visualized using ECL Western blotting detection reagents (Amersham Biosciences, Little Chalfont, Buckinghamshire, UK) and an LAS. Band intensity was measured using the Image-J program (NIH, Bethesda, MD, USA). Mouse anti-β –actin monoclonal antibody (1:20,000 dilution; Sigma-Aldrich, St. Louis, MO, USA), was used as a control for immunoblotting.

### 4.13. Statistics

Data are presented as mean ± SE. Statistical significance of the difference between groups was assessed using a two-sided student *t*-test. *p* < 0.05 was considered statistically significant.

## Figures and Tables

**Figure 1 ijms-20-05665-f001:**
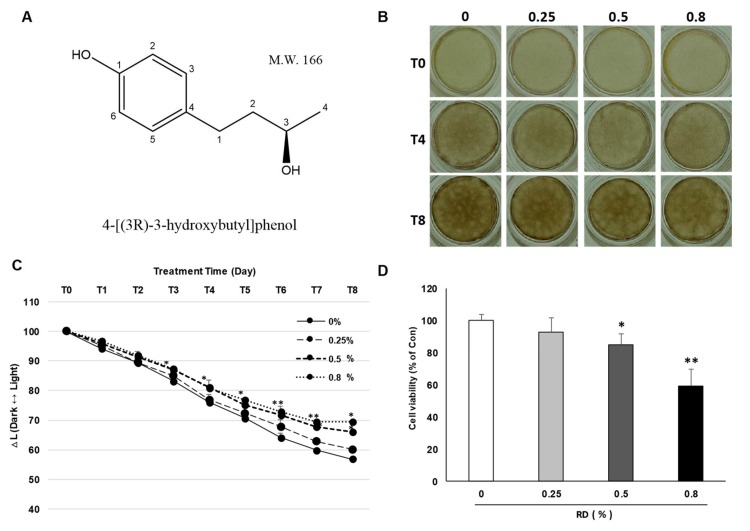
Effect of rhododenol (RD) on a pigmented 3D skin model, Melanoderm™. (**A**) Chemical structure of rhododenol; (**B**) color of 3D Melanoderm™; (**C**) the degree of hypopigmentation in Melanoderm™ as measured by the ∆L value in skin tissue between baseline and 17 days after RD or vehicle treatment; (**D**) cell viability was further confirmed in Melanoderm™, using WST-1 assay. Data are presented as the mean ± SE (*n* = 3). * *p* < 0.05 and ** *p* <0.01 versus DMSO control.

**Figure 2 ijms-20-05665-f002:**
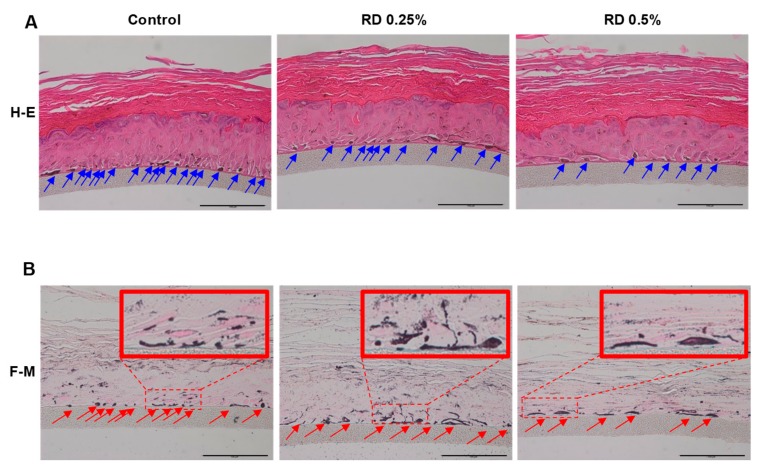
RD changes cell count and morphology of melanocytes in Melanoderm™. (**A**) Hematoxylin and eosin (H&E) stained Melanoderm™; (**B**) Fontana-Masson (F-M) stained tissues (scale bar= 100 μm).

**Figure 3 ijms-20-05665-f003:**
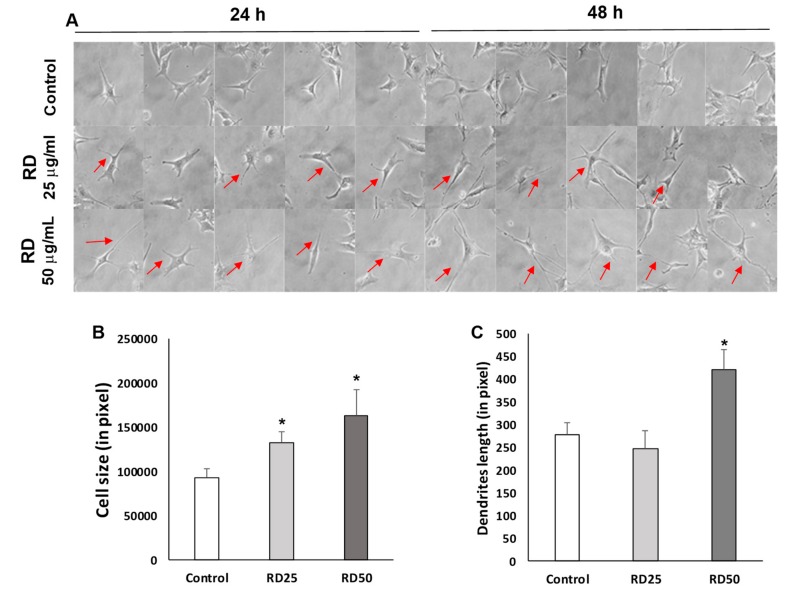
Increase of cell size and dendrite length by RD. (**A**) Visual assessment of B16F10 melanoma cells following the exposure to RD. Representative morphological changes were marked with red arrows. (400×) (**B**) Cell size changes 24 h after the exposure to RD. Polygonal contouring in ImageJ for measurement of cell size. Cell sizes in pixels. Data represent the mean ± SE of more than 6 cells (*n* ≥ 6). (**C**) Dendrites length (in pixel) changes 48 h after the exposure to RD. Straight lines in ImageJ for measurement of dendrites length. Dendrites length in pixels. Data represent the mean ± SE of more than 6 cells (*n* ≥ 6). * *p* < 0.05 versus DMSO control.

**Figure 4 ijms-20-05665-f004:**
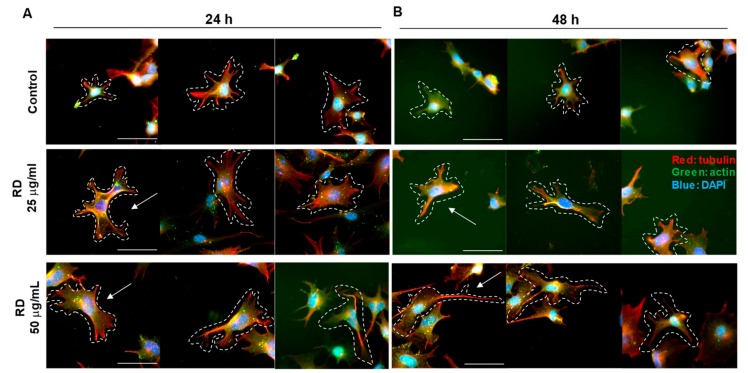
Ratio change of tubulin and actin by rhododenol. B16F10 cell were treated 0, 25, and 50 μg/mL RD for 24 h (**A**) and 48 h (**B**). β-actin is stained in B16F10 cells by immunocytochemistry. (detected with Alexa Fluor^®^ 488 Goat anti-Rabbit, shown in green); and Mouse monoclonal to Tubulin-Microtubule Marker (Alexa Fluor^®^ 594, shown in red). Nuclear DNA was labeled with 4,6-diamidino-2-phenylindole (DAPI) (shown in blue). Representative morphological changes were marked with white dotted lines and white arrows. The shape of the cells is indicated by dotted lines. The image was taken with a microscope (Nikon eclipse Ts2R, 400×, scale bar = 50 μm).

**Figure 5 ijms-20-05665-f005:**
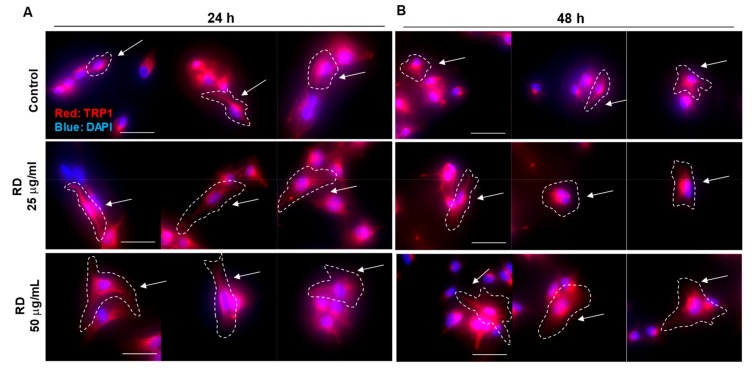
Increased expression of a melanosomal marker, TRP1. B16F10 cell were treated 0, 25, and 50 μg/mL RD for 24 h (**A**) and 48 h (**B**). TRP1 is stained in B16F10 cells by immunocytochemistry. (detected with Alexa Fluor^®^ 594 Goat anti-Mouse, shown in red). Nuclear DNA was labeled with DAPI (shown in blue). Representative morphological changes were marked with white dotted lines and white arrows. Image was taken with a microscope (Nikon Eclipse Ts2R, 400×, scale bar = 50 μm).

**Figure 6 ijms-20-05665-f006:**
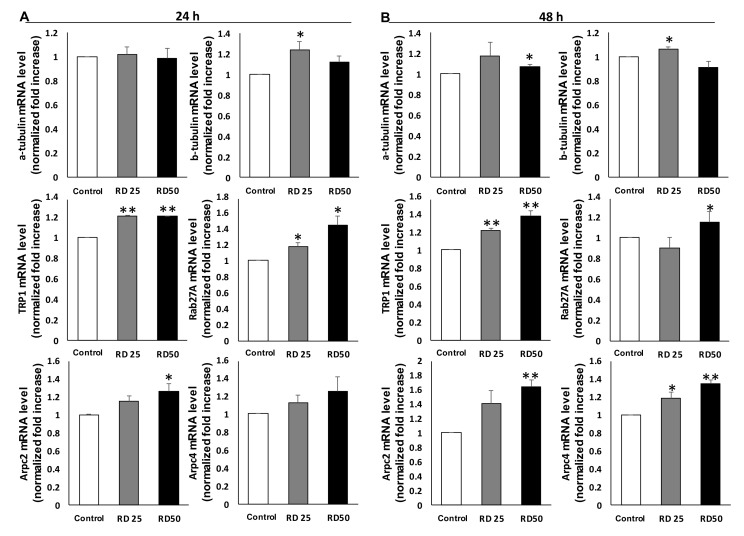
Expression of genes related to the cytoskeleton dynamics and melanization. Treated cells were subjected to real-time PCR (QPCR) for α tubulin, β tubulin, TRP1, Rab27A, Actin-related protein 2/3 sub2, and Actin-related protein 2/3 sub4. B16F10 cells were treated with RD at 0, 25, and 50 μg/mL for 24 h (**A**) and 48 h (**B**). Data are presented as the mean ± SE (*n* = 3, * *p* < 0.05 and ** *p* < 0.01, versus DMSO control).

**Figure 7 ijms-20-05665-f007:**
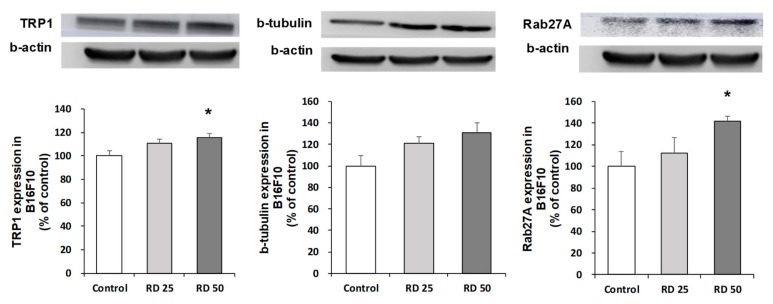
Effects of RD on the protein levels of TRP1, β-tubulin, and Rab27A. Treated cells were subjected to western blot analysis for TRP1, β tubulin, and Rab27A. B16F10 cells were treated with RD at 0, 25, and 50 μg/mL for 48 h. Data are presented as the mean ± SE (*n* = 3, * *p* < 0.05 versus DMSO control).

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
