# Peer review of "Rhododenol Activates Melanocytes and Induces Morphological Alteration at Sub-Cytotoxic Levels"

_ijms, 2019, doi:10.3390/ijms20225665_

Round 1

Reviewer 1 Report

In this manuscript entitled “Rhododenol Activates Melanocytes and Induces Morphological Alteration at Sub-Cytotoxic Levels”, there is some useful information, however, the further modification is required, and I have provided some advices shown below.

Major:

The dendrite formation and elongation of melanocytes helps to deliver melanin to keratinocytes and the skin to get dark. Why authors claim rhododenol induces the leukoderma from the opposite idea? Please add more discussion or experimental data of its mechanism. I recommend the author do western blotting to confirm the rhododenol regulate protein productions to verify the results of qRT-PCR.

Minor:

In Fig 4, the legend of a symbol is missed. Line 177, absent a comma. I suggest add published papers from IJMS for the references, such as “Silencing Stem Cell Factor Gene in Fibroblasts to Regulate Paracrine Factor Productions and Enhance c-Kit Expression in Melanocytes on Melanogenesis. DOI:10.3390/ijms19051475. ” and “Melanocyte Activation Mechanisms and Rational Therapeutic Treatments of Solar Lentigos DOI.org/10.3390/ijms20153666“.

Overall, this manuscript contains some useful information, and the discovery is significant. Including more data in the article to support discussion is necessary. To sum up, the manuscript should be in a major revision at this stage.

Reviewer 2 Report

Submitted to me for review article entitled “ Rhododenol Activates Melanocytes and Induces  Morphological Alteration at Sub-Cytotoxic Levels” is an original paper.

The authors discovered that rhododenol treatment reduced the number of melanocytes in a pigmented 3D human skin model, Melanoderm™, confirming the melanocyte toxicity of RD. In my opinion this interesting topic has been presented very clearly by the authors. In addition, the authors have experience in working with this compound, as evidenced by earlier publications (Kim, M.; Baek, H. S.; Lee, M.; Park, H.; Shin, S. S.; Choi, D. W.; Lim, K. M., 353 Rhododenol and raspberry ketone impair the normal proliferation of melanocytes through reactive oxygen species-dependent activation of GADD45. Toxicol In Vitro 355 2016,32, 339-46.). Currently a lot of research is focused on the natural biological active compounds to minimize toxicological side effect in patients. Abstract and Introduction  are well written, all elements are clearly presented, but in line 31 I suggest to add to the plant name, the name of authors (Acer nikoense and Betula platyphylla). The methodology is good performed and reasonably clear. Furthermore, the statistical methodology is appropriate. The analysis of the data is systematic and simple descriptive statistics, satisfactory.

The interpretation of the results is clearly presented and it is adequately supported by the evidence adduced. The conclusions are logically valid and justified by the evidence adduced. All figures are adequate and necessary, the photos are good quality. Although the discussion is short, it addresses all important aspects of the work. Despite interesting presentation on the topic I have a few minor comments which are submitted below:

Minor comments:- in the methodology section authors failed to explain where designed the primers used in the RT-PCR? - in the results section 2.2 The authors write about two concentrations of RD used in this studies. Why did the authors choose these concentrations? - in introduction and discussion please correct references because sometimes the authors write different [2], [3] should be [2,3].

Author Response

Q1. in line 31 I suggest to add to the plant name, the name of authors (Acer nikoense and Betula platyphylla).

Q2. Minor comments:- in the methodology section authors failed to explain where designed the primers used in the RT-PCR? - in the results section 2.2 The authors write about two concentrations of RD used in this studies. Why did the authors choose these concentrations? - in introduction and discussion please correct references because sometimes the authors write different [2], [3] should be [2,3].

A1 and A2. First of all, we thank the reviewer for favorable opinion on our study. We corrected all the Typo and added points raised by the reviewer in the revision.

The concentration of RD we used is sub-cytotoxic level,which we explained in the manuscript as shown below,

“Reflecting on previous mechanistic studies on RIL, cytotoxic levels of RD (³ 0.5 mM, ³ ~ 90 μg /mL in monolayer cells in vitro) [16] were mainly employed to study the toxicity of RD to melanocytes, but the effects of sub-cytotoxic levels of RD (£ 0.25 mM or £ 50 μg/mL) were not fully addressed.”

Round 2

Reviewer 1 Report

no

Author Response

thank you